# Axonal mechanisms mediating γ-aminobutyric acid receptor type A (GABA-A) inhibition of striatal dopamine release

Paul F Kramer[1], Emily L Twedell[1], Jung Hoon Shin[2], Renshu Zhang[1], Zayd M Khaliq[1]*

[1]Cellular Neurophysiology Unit, National Institute of Neurological Disorders and Stroke, National Institutes of Health, Bethesda, United States; [2]Laboratory on Neurobiology of Compulsive Behaviors, National Institute of Alcohol Abuse and Alcoholism, National Institutes of Health, Bethesda, United States

**Abstract** Axons of dopaminergic neurons innervate the striatum where they contribute to movement and reinforcement learning. Past work has shown that striatal GABA tonically inhibits dopamine release, but whether GABA-A receptors directly modulate transmission or act indirectly through circuit elements is unresolved. Here, we use whole-cell and perforated-patch recordings to test for GABA-A receptors on the main dopaminergic neuron axons and branching processes within the striatum of adult mice. Application of GABA depolarized axons, but also decreased the amplitude of axonal spikes, limited propagation and reduced striatal dopamine release. The mechanism of inhibition involved sodium channel inactivation and shunting. Lastly, we show the positive allosteric modulator diazepam enhanced GABA-A currents on dopaminergic axons and directly inhibited release, but also likely acts by reducing excitation from cholinergic interneurons. Thus, we reveal the mechanisms of GABA-A receptor modulation of dopamine release and provide new insights into the actions of benzodiazepines within the striatum.

**\*For correspondence:**
zayd.khaliq@nih.gov

**Competing interests:** The authors declare that no competing interests exist.

## Introduction

Axons of midbrain dopaminergic neurons are highly complex structures that transmit reward, associative-learning, and motor control signals to terminal boutons via action potentials that trigger the release of dopamine (*Aransay et al., 2015*; *Matsuda et al., 2009*; *Sulzer et al., 2016*). In addition to spike transmission, dopamine neuron axons within the striatum integrate local information. For example, striatal cholinergic interneurons modulate dopamine release through activation of nicotinic receptors on dopamine neuron axons (*Rice and Cragg, 2004*; *Zhang and Sulzer, 2004*) and synchronous activation of cholinergic interneurons can directly trigger dopamine release (*Cachope et al., 2012*; *Threlfell et al., 2012*). Similarly, other receptors have been shown to modulate dopamine release such as dopamine D2 (*Ford, 2014*), GABA-B (*Pitman et al., 2014*), metabotropic glutamate (*Zhang and Sulzer, 2003*), and muscarinic receptors (*Shin et al., 2015*). These data show that direct modulation of the axon presents a powerful means of controlling striatal dopaminergic signaling in a manner that is independent of somatic processing, suggesting a degree of functional segregation between these two cellular compartments (*Cachope and Cheer, 2014*; *Hamid et al., 2016*; *Mohebi et al., 2019*). Understanding the mechanisms that govern local control of dopamine release within the striatum will require better mechanistic knowledge of how presynaptic receptors shape axonal excitability.

GABA has long been known to modulate striatal dopamine release (*Giorguieff et al., 1978*; *Reimann et al., 1982*; *Starr, 1978*) but the specific contribution of GABA-A receptors (GABA-ARs) to this process is unclear. Fast-scanning cyclic voltammetry (FSCV) studies found that antagonists of GABA-ARs reduce dopamine release through an indirect mechanism involving $H_2O_2$ produced downstream of AMPA receptors, suggesting that GABA-ARs enhance dopamine release (*Avshalumov et al., 2003*; *Sidló et al., 2008*). By contrast, in vivo microdialysis studies have found that striatal infusions of GABA-AR antagonists lead to an increase in dopamine release, suggesting that striatal GABA-ARs inhibit dopamine release (*Gruen et al., 1992*; *Smolders et al., 1995*). Consistent with this finding, a recent FSCV study showed that GABA-AR activation leads to inhibition of dopamine release, but argued that the effect was indirect through GABA-B receptors located on dopamine neuron axons (*Brodnik et al., 2019*). A separate study showed that GABA-AR activation inhibits dopamine release in the absence of nicotinic receptor activation which led to the proposal that GABA-A receptors may be present on the terminals of dopaminergic neurons (*Lopes et al., 2019*). However, definitive evidence for this proposal is lacking.

Benzodiazepines are positive allosteric modulators of GABA-ARs that are increasingly prescribed in the United States (*Bachhuber et al., 2016*). These drugs have demonstrated misuse liability that in rare cases leads to a substance use disorder (*Blanco et al., 2018*). The mechanism of benzodiazepine reward is thought mainly to involve disinhibition of somatic firing (*Tan et al., 2010*). Similar to many drugs of abuse, systemically-applied benzodiazepines result in acute glutamate receptor plasticity in dopamine neurons (*Heikkinen et al., 2009*; *Kauer and Malenka, 2007*) and increase the frequency of individual dopamine release events in the striatum (*Schelp et al., 2018*). Unlike other drugs of abuse, however, benzodiazepines have been shown to decrease the amplitude of striatal dopamine release (*Gruen et al., 1992*; *Schelp et al., 2018*). These opposing effects suggest that benzodiazepines can differentially influence activity in the soma and release from axon terminals. To disentangle these conflicting results, we made direct axonal recordings from main axons and performed perforated-patch recordings of subthreshold voltage from branching processes of dopaminergic neuron axons within the striatum. In addition, we used calcium imaging, fast scanning cyclic voltammetry, and fluorescent sensor imaging of dopamine release. Our experimental findings provide mechanistic understanding of how axonal GABA-A receptors control the excitability and transmitter release from dopamine neuron axons.

## Results

### Characteristics of action potentials in dopaminergic neuron axons – main axons and striatal terminal axons

Dopamine neurons of substantia nigra *pars compacta* form thin, unmyelinated axons that project to the dorsal striatum through the medial fiber bundle (MFB). To examine action potential firing in the main unbranching axon from adult mice, we used a horizontal brain slice preparation which preserved the connection between the cell bodies of SNc dopaminergic neurons and their MFB projecting axons. Dopaminergic neuron axons within the MFB were identified using the fluorescent marker proteins GFP or td-Tomato from TH-GFP or DAT-CRE x Ai9 mice, respectively. Using these optimized brain slices in combination with marker mice enabled us to record propagating action potentials from the main axon at distances of greater than two millimeters from the soma (*Figure 1A*).

To examine the characteristics of axonal action potentials, we made whole-cell recordings from the cut ends of axons (blebs) located on the surface of the slice (*Hu and Shu, 2012*; *Hu et al., 2009*; *Shu et al., 2007*). We found that many axons exhibited spontaneous firing activity with median spontaneous rates that were nearly identical to somatic pacemaker rates (*Figure 1B*; axon: $\tilde{x}$ = 3.3 Hz, n=41; soma: $\tilde{x}$ = 2.75 Hz, n=10; Mann-Whitney U test, $U$=161, p=0.298), consistent with the slow, rhythmic firing associated with dopaminergic neurons (*Grace and Bunney, 1984*). Axonal action potentials had narrower half-widths (*Figure 1C and D*; axon: $\tilde{x}$ = 0.89 ms n=27, soma: $\tilde{x}$ = 1.24 ms n=10; Mann-Whitney $U$=59, p=0.008 two-tailed) and a more hyperpolarized threshold relative to somatic spikes (*Figure 1E*; axon: $\tilde{x}$ = -56 mV, n=26; soma: $\tilde{x}$ = -41.7 mV n=10; $U$=9, p<0.0001).

In the axon, the voltage trajectory between action potentials was shallow in slope (avg. dV/dt at middle 50% of the interspike interval, axon: $\tilde{x}$ = 9.24 mV/s, n=27), reaching a minimum at the spike

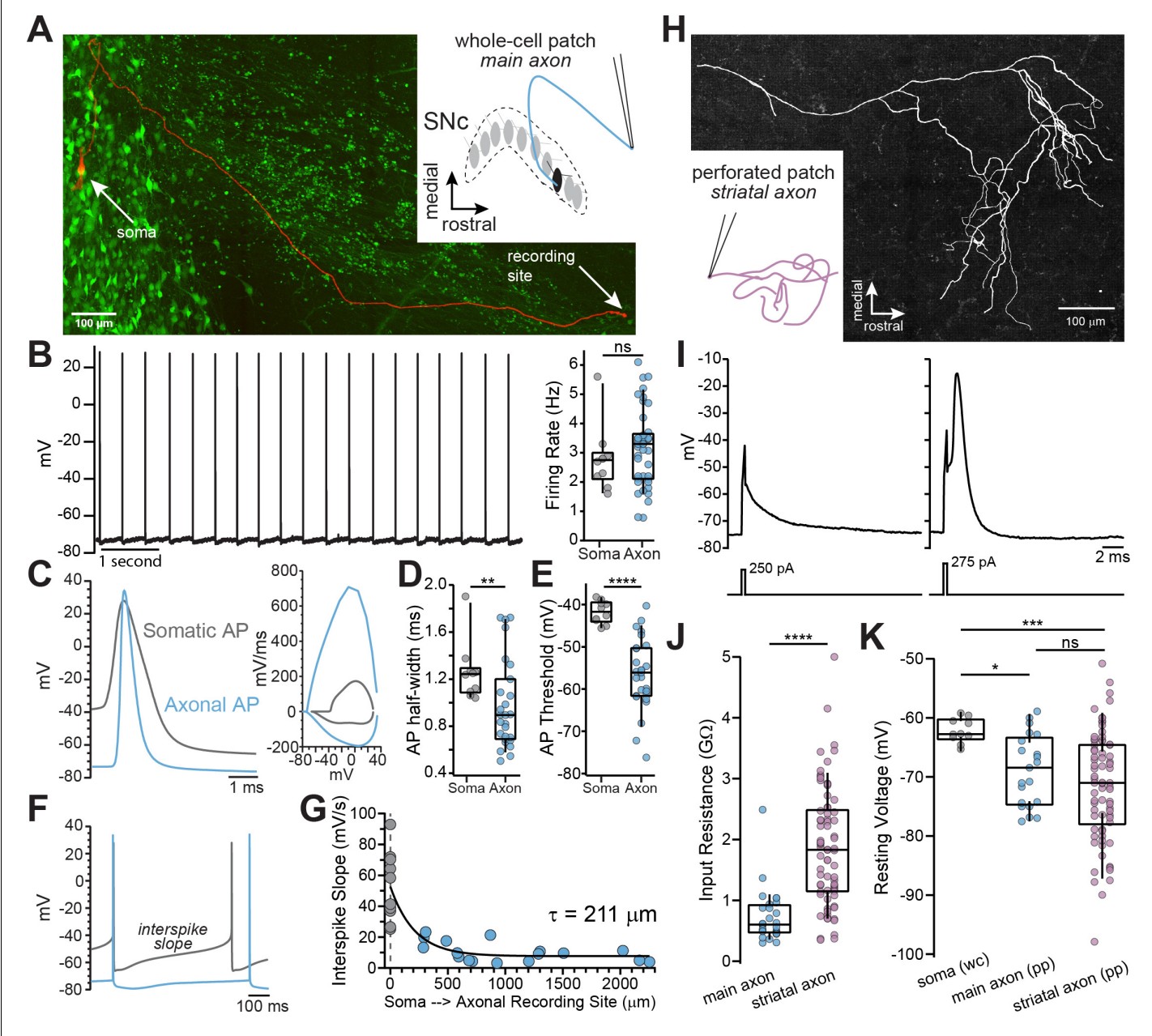

**Figure 1.** Whole-cell and perforated-patch recordings from dopamine neuron axons. (A) Main axon recorded in whole-cell mode with a connected soma (filled with neurobiotin, imaged with streptavidin-Cy5, slice cleared with CUBIC, *red*); GFP driven by the TH promoter (*green*). (B) Trace of spontaneous action potentials recorded from a dopaminergic axon (*left*). Firing rate from somatic (*n* = 10) and axonal recordings (*n* = 41; p=0.298) (*right*). (C) Overlay of an axonal and somatic spike (*left*). Phase plot for a somatic and axonal action potential (*right*). (D) Half-peak widths from somatic (*n* = 10) and axonal (*n* = 27) APs (**p=0.008). (E) AP thresholds recorded from soma (*n* = 10) and axons (*n* = 26)(****p<0.0001). (F) Example traces of interspike voltage from obtained from axonal (*blue*) and somatic (*gray*) recordings. (G) Slope of interspike voltage plotted against recording distance between axonal recording site (*blue*) and soma (*gray*). (H) Post-hoc reconstruction of a patched striatal axon. (I) Trace of subthreshold depolarization (*left*) and axonal AP (*right*) evoked by 250 pA and 275 pA current injection. (J) Input resistance values for main axon (*n* = 28) and striatal axons (*n* = 74) APs (****p<0.0001). (K) Comparison of the mean interspike voltage between soma (*n* = 10) main axon (*n* = 21) and striatal axon, which was measured as the average resting membrane potential (*n* = 74) (*p=0.032; ***p=0.0007; ns p=0.87).

The online version of this article includes the following video and source data for figure 1:

**Source data 1.** Raw values used for plots in *Figure 1*.
**Figure 1—video 1.** Animated rotating movie of a striatal filled axon.
https://elifesciences.org/articles/55729#fig1video1

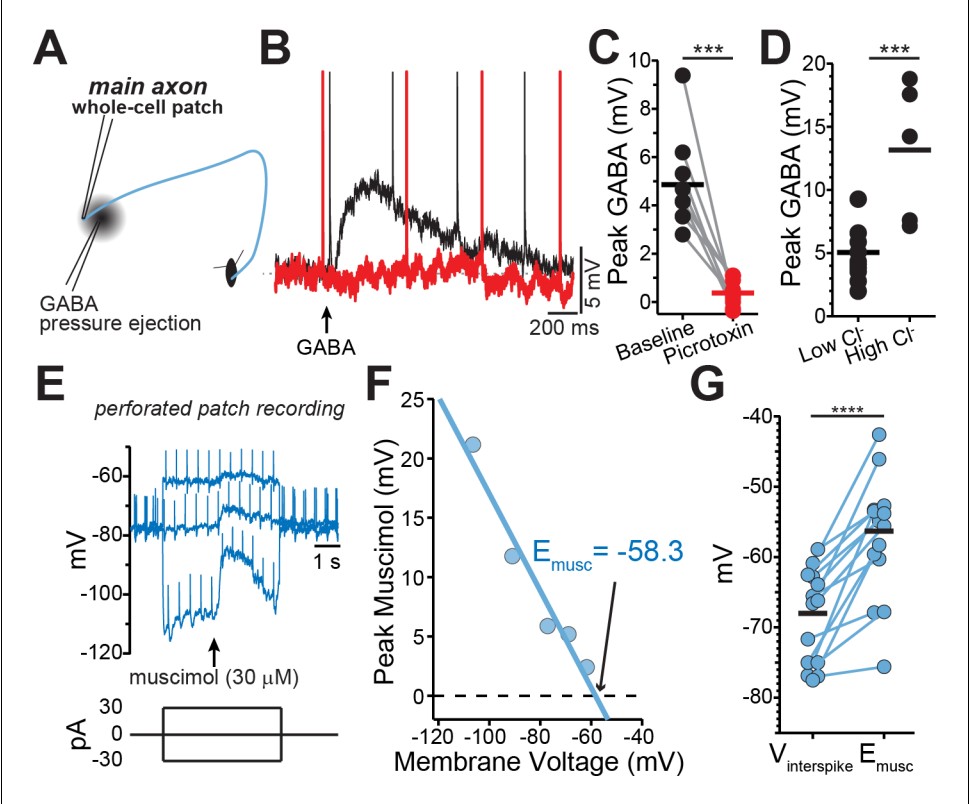

**Figure 2.** GABA-A receptors located on the axons of dopaminergic neuron are depolarizing. (A) Schematic of experimental setup. (B) Example traces of pressure ejection of GABA onto the axon in control (*black*) and after application of picrotoxin (*red*). (C) Peak of amplitude of GABA-evoked depolarization in control and after picrotoxin ($n = 9$; ***p=0.0003). (D) Peak amplitude of GABA-evoked depolarization recorded with low ($n = 5$) or high ($n = 10$) chloride internal solutions (***p=0.0008). (E) Example traces overlaid from three different current injections. Timing of muscimol puff application indicated by *arrow*. (F) Peak of the muscimol-evoked depolarization plotted against the average interspike voltage. *Fitted line* was used to determine the reversal potential. (G) Paired values indicating mean interspike voltage ($V_{interspike}$) and muscimol reversal potential ($E_{musc}$) in individual axons ($n = 15$; ****p<0.0001).

The online version of this article includes the following source data for figure 2:

**Source data 1.** Raw values used for plots in *Figure 2*.

trough with little depolarization before reaching threshold. By contrast, the somatic interspike voltage exhibited a significantly greater slope on average (*Figure 1F*; avg. dV/dt, soma: $\tilde{x} = 49.9$ mV/s n=10; $U$=22, p<0.0001), similar to previously reported values (*Khaliq and Bean, 2008*). A plot of the slope of the interspike voltage against the axonal recording distance followed a roughly exponential relationship with the interspike slope, such that it becomes more shallow with increasing recording distance (*Figure 1G*; single exponential fit, length constant, λ=211 μm, n=27; $R^2$=0.70; data were fit with a single exponential significantly better than with a line: F(1,24)=22.1, p<0.0001). Axonal recordings at distances greater than two length constants from the soma (> 422 μm) exhibited little sub-threshold depolarization between action potentials ($\tilde{x}$=7.3 mV/s, n=13). In sum, action potentials recorded in the main axons of dopaminergic neurons are narrow, with voltage thresholds that are negative relative to somatic spikes.

Within the dorsal striatum the axons of dopaminergic neurons branch extensively while progressively decreasing in diameter (*Matsuda et al., 2009*), which raises the question of how the properties of striatal terminal axons compare to those of the main axon. Using perforated-patch recordings to record from axon blebs, we found that terminal axons have a higher input resistance (*Figure 1J*; $\tilde{x} = 1.83$ GΩ, n=74) than the main axon ($\tilde{x} = 599$ MΩ, n=28, $U$=254, p<0.0001 two-tailed). The interspike membrane potential in the striatal dopamine neuron axon was hyperpolarized relative to the main axon, but both axonal compartments were more hyperpolarized relative to the average

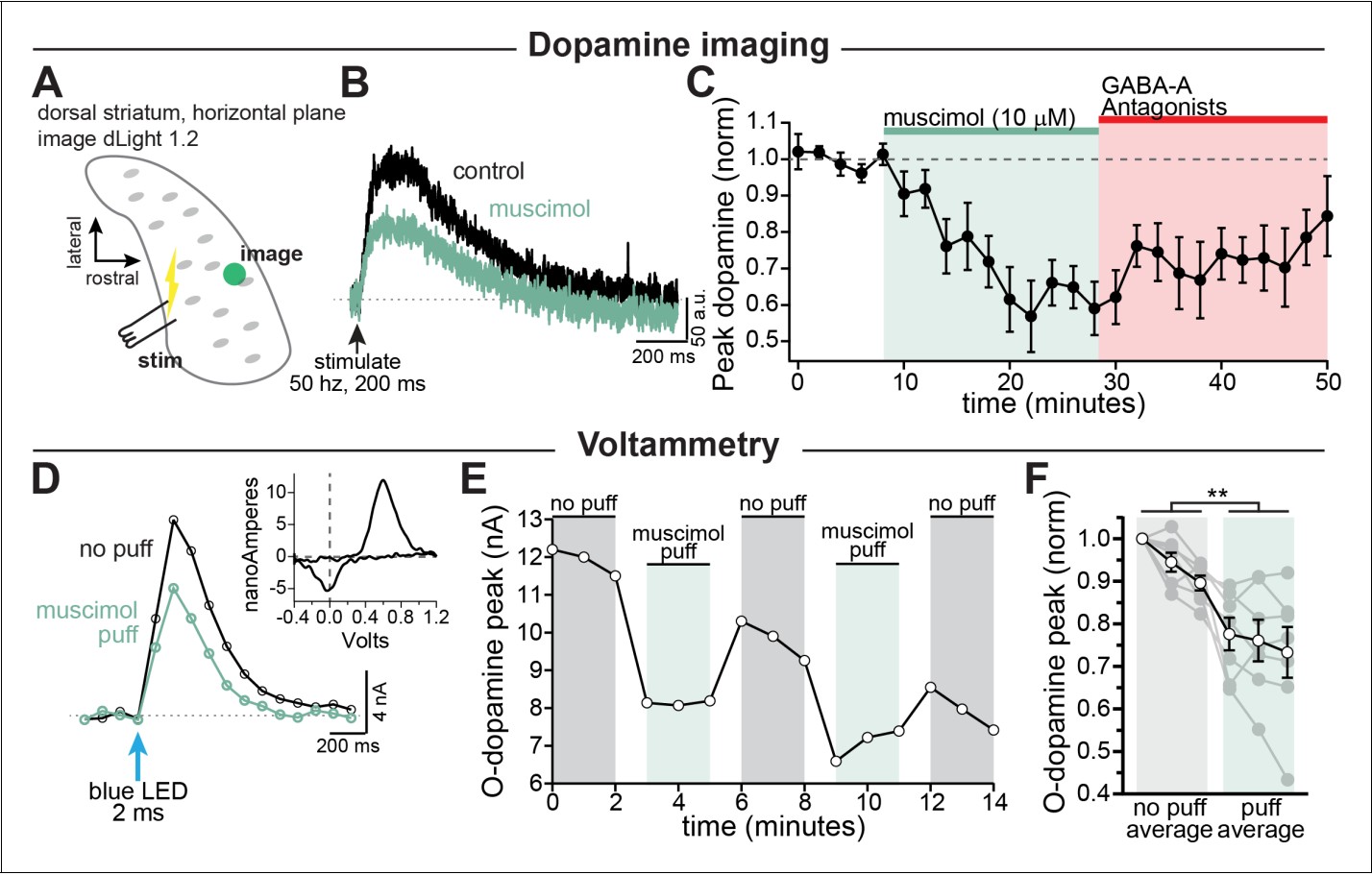

**Figure 3.** GABA-A receptor mediated inhibition of striatal dopamine release. (**A–C**) Imaging dopamine release in the dorsal medial striatum using dLight 1.2. (**A**) Experimental setup diagram. (**B**) Example traces of bulk dopamine release following electrical stimulation in control (black) and during muscimol bath application (green). (**C**) Group data showing the time course of muscimol application on dopamine release peak amplitude measured distal to the site of stimulation (average 1054 μm, n = 9; p=0.0008). (**D–F**) Voltammetry experiments using opsin-evoked dopamine release (DAT-Cre injected with CoChR in SNc). (**D**) Example voltammetry signal of dopamine in control (*black*) and during muscimol puff (*green*). Inset: example current-voltage plot of FSCV signal. (**E**) Time course of opsin-evoked dopamine release in an example experiment. (**F**) Summary data showing the main effect of muscimol puff on dopamine release measured with voltammetry. *Closed symbols* indicate individual experiments and *open symbols* indicate averaged data (p=0.004; n = 7). All experiments in (**A–C**) were done in the presence of hexamethonium chloride (200 μM), sulpiride (1 μM), atropine (30 nM), and CGP 55845 (200 nM) to block nAChRs, D2Rs, mAChRs, and GABA-BRs respectively.

The online version of this article includes the following source data and figure supplement(s) for figure 3:

**Source data 1.** Raw values used for plots in *Figure 3* and *Figure 3—figure supplement 1*.
**Figure supplement 1.** GABA-A acitvation does not affect baseline dLight levels.

interspike voltage in the soma (*Figure 1K*; terminal axon: $\tilde{x}$ = -71 mV, main axon: $\tilde{x}$ = -68.9 mV, soma=-62.2 mV; Kruskal-Wallis H test $\chi^2$(2)=13.9, p=0.001; terminal vs. main p=0.87; terminal vs. soma, p=0.0007; soma vs. main, p=0.032). Together, these results show that the main and terminal axons of dopamine neurons are high input resistance compartments in which action potentials are evoked from relatively hyperpolarized interspike voltages.

## Identification of GABA-A receptor-mediated currents on dopaminergic neuron axons

Past work has shown that GABA-A receptors modulate dopamine release, but evidence that GABA-A receptors are located on dopaminergic neuron axons has been indirect. To test for a GABA-A receptor-mediated conductance in the axon, a second pipette was placed 30–60 μm from the axonal recording site on the main axon and GABA (300 μM-1 mM) was locally applied by a brief (80–300 ms) pressure ejection (*Figure 2A,B*). GABA puff resulted in depolarization of the axonal

membrane potential by an average of 4.86 ± 0.66 mV (n = 9), which was completely blocked by the GABA-A antagonist picrotoxin (*Figure 2C*; 100 μM; t(8)=6.1, p=0.0003). To verify the direct nature of these currents, we tested the effect of increasing the concentration of intracellular chloride on the GABA-mediated depolarization. We found that filling axons with an internal solution containing high chloride resulted in GABA-mediated depolarizations that were 2.76-fold larger in amplitude (*Figure 2D*; low Cl⁻=4.74 ± 0.66 mV, high Cl⁻=13.1 ± 2.44 mV; t(13)=4.34, p=0.0008). These results provide direct evidence for the presence of functional GABA-A receptors on the axons dopaminergic neurons.

## Axonal GABA-A receptors are depolarizing

The physiological function of GABA-A receptors is closely tied to its reversal potential, which has been shown to vary in axons across cell types from depolarizing (*Pugh and Jahr, 2011*; *Ruiz et al., 2010*; *Szabadics et al., 2006*) to hyperpolarizing (*Rinetti-Vargas et al., 2017*; *Xia et al., 2014*). Therefore, we determined the GABA-A reversal potential in the main dopaminergic neuron axons using perforated-patch recordings in which the intracellular chloride concentration is preserved. While holding the axon at different membrane voltages with constant current, we applied single puffs of the GABA-A selective agonist muscimol and then measured the amplitude of the resulting muscimol-evoked membrane depolarization (*Figure 2E–G*). Our analysis showed that the average GABA-A reversal potential in dopamine neuron axons was −56.3 ± 2.38 mV (n = 15). Importantly, we found in all recorded axons that the reversal potential of axonal GABA-A current was always depolarized relative to the average interspike voltage of the axon (*Figure 2E–G*; $V_{interspike}$=-68 ± 1.75 mV, p<0.0001).

## GABA-A activation inhibits striatal dopamine release

Based on our finding that GABA-A receptors are depolarizing, we reasoned that activation of axonal GABA-receptors should enhance dopamine release. Therefore, we tested the effect of axonal GABA-A receptors on dopamine release using dLight 1.2 to image striatal dopamine. A stimulating electrode was placed at the caudal end of the striatum, and a burst of stimulations were elicited with a bipolar electrode (*Figure 3A–C*). Contrary to our expectation, we found that bath perfusion of muscimol (10 μM) depressed dopamine release by an average of 38.2 ± 6.6% (*Figure 3C*; RM 1-way ANOVA F(2, 26)=16.5; Bonferroni's post-hoc t(8)=5.8, p=0.0008). While we observed a large inhibition in evoked dopamine release, we saw no change in the pre-stimulation (baseline) dLight signal following muscimol perfusion suggesting that muscimol does not alter background striatal dopamine (*Figure 3—figure supplement 1*).

Past work on parallel fibers in the cerebellum has shown that GABA produces biphasic effects on axonal excitability with low intensity GABA uncaging (*Khatri et al., 2019*) or transient GABA-A receptor activation within a 10-20 second time window being more likely to enhance release while stronger or longer muscimol application being more likely to inhibit release (*Stell, 2011*). To test this hypothesis on dopamine neuron axons, we activated GABA-A receptors transiently with a puffer pipette (10 μM, 1-3 s puff). Dopamine release was evoked selectively from dopaminergic fibers in dorsal striatum slices using optical activation of a channelrhodopsin variant, CoChR. Extracellular dopamine was monitored using fast-scan cyclic voltammetry. Surprisingly, we found that puff application of muscimol still resulted in inhibition of dopamine release from axons within the dorsal striatum by an average of 19.1 ± 4.2% (*Figure 3D–F*; control, $\bar{x}$=94.7%; muscimol, $\bar{x}$=75.7%; F(1, 6)=20.7, p=0.004; n=7 slices). Together, these results show that both transient and steady-state receptor activation of axonal GABA-A receptors results in inhibition of striatal dopamine release.

## Axonal GABA-A receptors inhibit through a combination of voltage-gated sodium channel inactivation and shunting

To better understand how axonal GABA-A receptors inhibit dopamine release, we tested the effect of GABA-A receptor activation on axonal action potential waveforms. As shown in *Figure 4B*, the most prominent effect of GABA-A receptor activation was a shortening of the action potential peak. Although the effect of GABA on spike height varied between axons, we found that the peak was shortened on average by 7.74 ± 1.83 mV (avg. peak amplitude; control, $\bar{x}$ = 12.5 ± 4.72 mV; GABA, $\bar{x}$ = 4.77 ± 5.22 mV; 2-way ANOVA Bonferroni's post-hoc t(12)=5.75; n=7, p=0.0002). This effect was

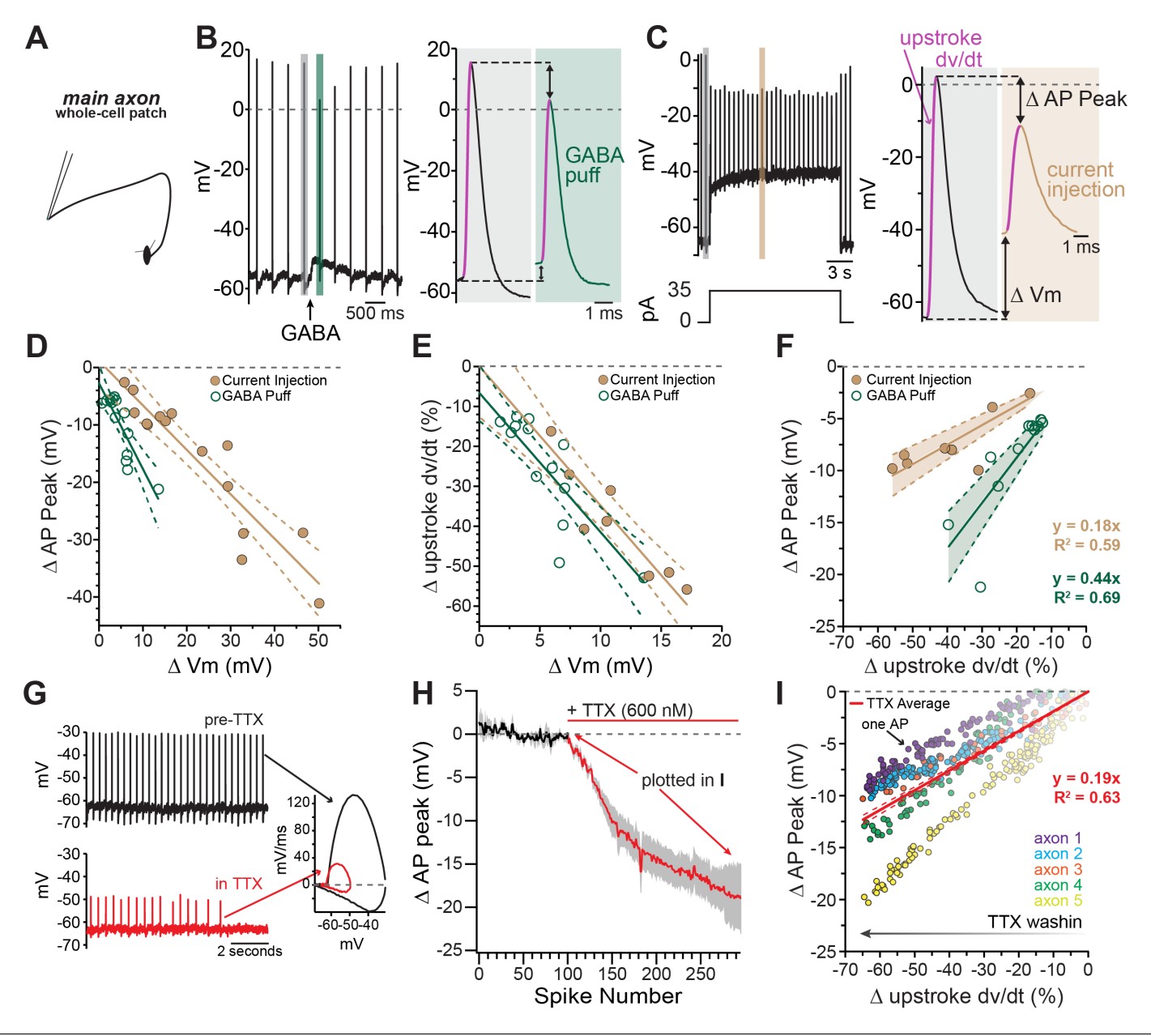

**Figure 4.** Axonal GABA-A receptors inhibit excitability through a combination of sodium channel inactivation and shunting. (**A**) Schematic of the experimental setup for this figure. Whole-cell recordings were made from the main, connected, axons of dopamine neurons. (**B**) Example axonal recording showing the effect of a brief GABA pressure ejection. Control (gray) and GABA (green) traces magnified. *B*, *right*: Control action potential in black and GABA-affected action potential in green. (**C**) Similar experiment to B, except demonstrating the effect current injection on spike properties as opposed to GABA application. Purple line on AP upstroke denotes area of measurement for rate of rise, arrows denoting measurement of change in AP peak and change in membrane potential. (**D**) Effect of the amount of baseline depolarization on the decrease in peak AP amplitude, compared between GABA (green; n = 14) and current injection (tan; n = 15) (*p=0.047). (**E**) Effect of the amount of baseline depolarization on the normalized decrease in rate of AP upstroke, compared between GABA (green; n = 11) and current injection (tan; n = 10) (ns p=0.564). (**F**) A plot showing the relationship between decrease in rate of AP rise and decrease in AP peak, for injection of current (brown; n = 8) and a brief pressure ejection of GABA (green; n = 11). (**G**) Example axonal recording showing spontaneously firing action potentials before the application of TTX (*top, black*) and after TTX bath perfusion, just before the action potentials cease (*bottom, red*), *inset*: Example phase plots for axonal action potentials before (black) and after (red) TTX perfusion. (**H**) Averaged data showing the effect of TTX on action potential peak amplitude (graphs were aligned to the beginning of TTX effect, n = 5). The decrease in peak amplitude is plotted in I. (**I**) Data from five individual axons showing the effect of TTX wash-in on the change in rate of action potential rise, and the change in the peak of the action potential. Each dot in data from an individual action potential, normalized to just before the perfusion of TTX. In red is the average effect.

*Figure 4 continued on next page*

*Figure 4 continued*

The online version of this article includes the following source data for figure 4:

**Source data 1.** Raw values used for plots in *Figure 4*.

blocked completely by picrotoxin (peak reduction in picrotoxin; control peak, $\bar{x}$ = -1.86 ± 5.34 mV, GABA peak $\bar{x}$ = -3.88 ± 5.02 mV; 2-way ANOVA Bonferroni's post-hoc t(12)=1.49; n=7, p=0.32). We took advantage of the variability between axons in their responses to GABA in order to assess the relationship between the GABA-A mediated depolarization and spike height. Plotting data from 14 axon recordings, we found that the reduction in spike height correlated linearly with the GABA-mediated depolarization with a slope of -1.50 mV/mV (95% CI: -2.05 to -0.94; $R^2$=0.74, n=14; *Figure 4D*, fit to green symbols). Therefore, larger GABA-induced subthreshold depolarizations result in shorter axonal action potentials.

The GABA-A mediated reduction of spike amplitude likely involves two main processes: inactivation of axonal sodium channels due to GABA-induced depolarization (*Rama et al., 2015*) and shunting inhibition (*Xia et al., 2014*; *Cattaert and El Manira, 1999*). To dissect the contribution from these two processes, we compared the GABA puff experiments in *Figure 4B* to separate experiments where depolarization was evoked instead by direct current injection (*Figure 4C*). We reasoned that the effects of current injection-evoked depolarization on spike height should be dominated by sodium channel inactivation, whereas shunting inhibition should be minimal under these conditions. Plotting the spike height against current injection-evoked depolarization in *Figure 4D* (brown symbols), we found that direct current injections were significantly less effective at reducing spike peak amplitudes as compared to GABA mediated depolarization, shown by a shallower slope of best-fit lines (*Figure 4D*; GABA-A activation: −1.50 mV/mV, 95% CI: −2.05 to −0.94; direct depolarization: −0.77 mV/mV, 95% CI: −0.96 to −0.59; F(1,25)=4.39, n = 29, p=0.047). We next analyzed the rate of the rise of the action potential (dV/dt) as it reflects the maximal spike-evoked sodium current. By contrast, we found little difference in the effect of GABA-evoked and direct current injection-evoked depolarization on the rate of rise of axonal action potentials (dV/dt). Plots in *Figure 4E* show that both manipulations slowed the rate of rise of action potentials and shared similar dependences on subthreshold depolarization (slope of linear fits; GABA-A activation: −3.78 %/mV 95% CI: −5.46 to −2.10, direct depolarization: −3.20 %/mV, 95% CI: −4.57 to −1.83; F (1,17) = 0.35, n = 21, p=0.56).

These data suggested the peak amplitude of the action potential was susceptible to both depolarizations and shunting inhibition, while the rate of rise was only affected by depolarizations. In order to combine these two effects and distinguish between shunting inhibition and depolarization-mediated inactivation of sodium channels, the change in rate of rise was graphed against the change in peak spike amplitude. From this relationship the added effect of shunting inhibition is clear in the significantly steeper relationship for GABA-A receptor activation (*Figure 4F* slope of linear fits; GABA-A activation: 0.44 mV/%, 95% CI: −0.35 to 0.52, direct depolarization: 0.18 mV/mV, 95% CI: 0.15 to 0.22; F (1,17)=39.9, n = 19, p<0.0001).

To experimentally test the effect of sodium channel inhibition on axonal action potentials, TTX was bath perfused while recording axonal action potentials (*Figure 4G*). As the effect of TTX developed, the amplitude of the peak of the action potential was progressively reduced, and the rate of rise was progressively slowed (*Figure 4G–I*). We compared the relationship of the reduction in the peak and the slowing of the rate of rise across groups and found that the average of the TTX condition was similar to the direct depolarization, indicating this effect was mainly through inactivation of sodium channels. However, GABA-A receptor activation had a significantly steeper relationship, revealing the additional contribution of shunting inhibition (*Figure 4F* and *Figure 4I*).

These data show that GABA-A receptors act mechanistically through both depolarization of the axonal membrane as well as a change in the input resistance that leads to shunting inhibition. These two effects combine to slow and shorten dopaminergic action potentials.

## GABA-A receptor-mediated inhibition of axonal calcium signals increases with propagation distance

Our data show a clear inhibitory effect of axonal GABA-A conductance on dopamine release that differs from the excitatory actions of GABA-A receptors observed in most central axons. As a possible explanation for their distinctive responses, dopamine neuron axons in the dorsal striatum are distinguished by their very thin diameters and highly branched structure (*Matsuda et al., 2009*). Under some circumstances, this feature may present a challenge for spike propagation following the GABA-A increase in shunting inhibition.

We therefore set out to determine the influence of GABA-A receptors on signals that have propagated through the extreme architecture of the dopaminergic neuron terminals. To test this, a burst of activity was evoked in axons using a bipolar electrode placed in the caudal striatum (closer to the main trunk of the axon) and axonal calcium signals were imaged distal to the site of stimulation in the rostral striatum ($\bar{x} \sim 690 \pm 73.1$ µm, *Figure 5A–C*). We found that calcium signals evoked by electrical stimulation in the caudal striatum and imaged distally in the rostral striatum were significantly inhibited by GABA-A receptor activation (*Figure 5C*, muscimol $\bar{x}$=36.4 ± 14.3% of baseline; Šidák's post-hoc test t(4)=4.4; p=0.034; n=5 slices). By contrast, calcium signals evoked by local stimulation ($\bar{x} \sim 100 \pm 29.2$ µm between stimulator and imaging site, *Figure 5D–I*) were not significantly affected by bath perfusion of muscimol (10 µM). This was true for axonal calcium signals evoked by local stimulation of the caudal striatum (*Figure 5F*, muscimol $\bar{x}$=95.0 ± 8.4% of baseline; Šidák's post-hoc test t(4)=0.52; p=0.95; n=5 slices) and the rostral striatum (*Figure 5I*, muscimol $\bar{x}$=96.1 ± 3.2% of baseline; Šidák's post-hoc test t(7)=1.2; p=0.61; n=8 slices). Therefore, these data show that calcium signals in dopaminergic neuron axons of the striatum are most greatly affected by GABA-A receptor activation under conditions involving action potentials that propagate long distances throughout the axonal arbor.

## Benzodiazepines enhance tonic GABA activity on dopamine neuron axons

Benzodiazepines constitute a class of allosteric modulators that act on GABA-A receptors to enhance GABA-mediated currents. Much is known about the somatic mechanisms regulating the effects of benzodiazepines in dopamine neurons (*Reynolds et al., 2012*; *Tan et al., 2010*; *Tan et al., 2011*); but less is known about direct axonal effects of these drugs. Studies examining the effect of benzodiazepines on dopamine release showed these effects are likely mediated through GABA-B receptors, and indirectly involved GABA-A receptors on non-dopaminergic neurons (*Brodnik et al., 2019*). We therefore sought to determine the contribution of GABA-A receptors on dopamine neuron axons to the effects of diazepam.

We first tested the effect to inhibiting both GABA-A and GABA-B receptors in the striatum in the absence of any exogenously applied agonist. We found that co-application of GABA-A (gabazine) and GABA-B (CGP-55845) antagonists significantly enhanced dopamine release to 115% of baseline (*Figure 6—figure supplement 1A and B*; t(9)=2.99, p=0.015, n = 10) while gabazine alone did not significantly increase dopamine release (*Figure 6—figure supplement 1C and D*). These results are consistent with previous studies reporting a GABA tone in the striatum (*Ade et al., 2008*; *Gruen et al., 1992*; *Lopes et al., 2019*).

Next, we tested the effect of the benzodiazepine diazepam on modulating axonal GABA-A receptors. To do this, we used electrical stimulation to evoke dopamine release which we imaged using dLight. Experiments were performd in a cocktail of blockers that included antagonists against dopamine D2, GABA-B, and muscarinic. We found that diazepam robustly decreased dopamine release (*Figure 6A*; diazepam: 50.7 ± 11.3% of baseline; RM 1-way ANOVA Bonferroni's post-hoc t(4)=4.37; p=0.024, n = 5 slices). Under these conditions, a substantial amount of dopamine release is likely driven through cholinergic interneurons activation of nicotinic receptors (*Cachope and Cheer, 2014*; *Rice and Cragg, 2004*; *Threlfell et al., 2012*; *Zhang and Sulzer, 2004*). In order to isolate the diazepam effect due to direct stimulation of dopaminergic neuron axons, we imaged in a cocktail of synaptic blockers that also inhibited nicotinic receptors. In the presence of nicotinic receptors blocker, we found that diazepam application led to a weaker but still substantial inhibition of dopamine release (*Figure 6B*; diazepam: 80.6 ± 5.9% of baseline; RM 1-way ANOVA Bonferroni's test t(5)=3.28, n = 6 slices; p=0.044). Therefore, diazepam produces a sizeable inhibition of striatal

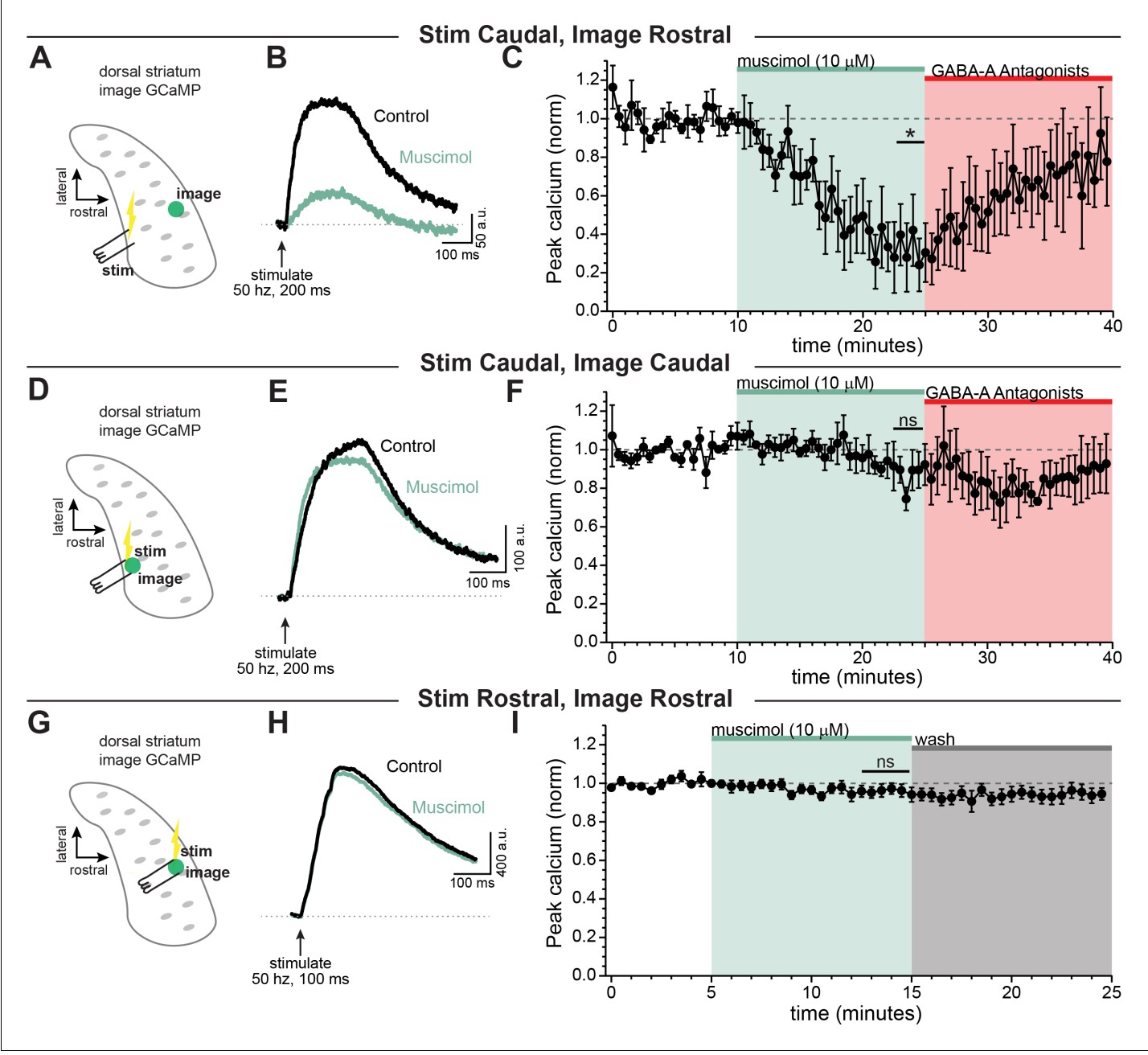

**Figure 5.** GABA-A receptor-mediated inhibition of axonal calcium signals increases with propagation distance. Stimulation-evoked calcium signals in dopamine neuron axons imaged either at the site of stimulation, or distal to the site of stimulation. (A) Experimental setup diagram. (B) Stimulation-evoked GCaMP6f signals in control (*black*) and in response to muscimol bath application (*green*). (C) Time course showing a significant effect of bath applied muscimol for GCaMP6f signals imaged in the rostral striatum, stimulated caudally (~690 ± 73.1 μm between stimulation and imaging, *n* = 5; *p=0.034). (D) Experimental setup diagram. E) Stimulation-evoked GCaMP6f signals in control (*black*) and in response to muscimol bath application (*green*). (F) Time course showing no significant effect of bath applied muscimol for GCaMP6f signals imaged near to the site of stimulation in the caudal striatum (~100 ± 29.2 μm between stimulation and recording, *n* = 5, p=0.95). (G) Experimental setup diagram. H) Stimulation-evoked GCaMP6f signals in control (*black*) and in response to muscimol bath application (*green*). (I) Time course showing no significant effect of bath applied muscimol for GCaMP6f signals imaged near to the site of stimulation in the rostral striatum (<100 μm between stimulation and recording, *n* = 8, p=0.61). In these experiments, Hexamethonium Chloride (200 μM; *Figure 6A–F*) or DhβE (1 μM; *Figure 6G–I*), sulpiride (1 μM), atropine (30 nM), and CGP 55845 (200 nM) were used to block nAChRs, D2Rs, mAChRs, and GABA-BRs respectively.

The online version of this article includes the following source data for figure 5:

**Source data 1.** Raw values used for plots in *Figure 5*.

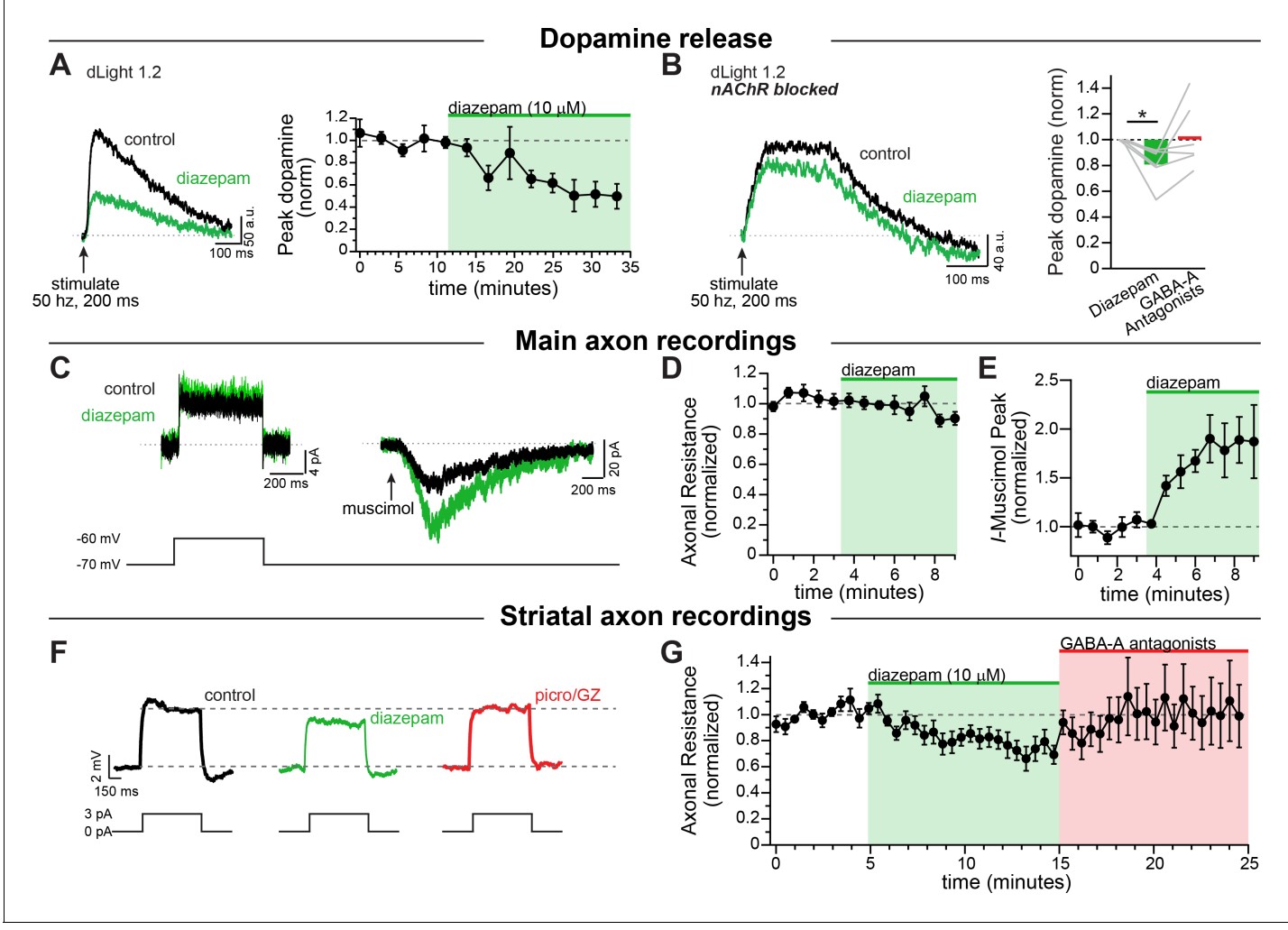

**Figure 6.** Diazepam inhibits striatal dopamine release through direct effects on axonal GABA-A receptors. (A) Example traces of imaged dopamine release from control (black) and diazepam bath perfusion (green) conditions. *Right:* time course showing the effect of diazepam bath perfusion on peak dopamine release (*n* = 5). (B) Example traces of imaged dopamine release from control (black) and diazepam bath perfusion (green) conditions with nAChRs blocked. *Right:* Group effect of diazepam on peak dopamine release (*n* = 6; *p=0.044). GABA-A Antagonists picrotoxin (100 μM) and gabazine (10 μM) were perfused at the end of each experiment. (C) Example step depolarization (*left*) and muscimol pressure ejection (*right*) recorded in the main axon in control (black) and diazepam bath application (green). Step depolarization and muscimol puff were performed within the same cell. (D) Time course showing the effect of diazepam bath application on the normalized input resistance of the main axon in the medial forebrain bundle (*n* = 5). (E) Time course showing the effect of diazepam bath application on the normalized muscimol-evoked peak current (*n* = 6). (F) Example current injections to test axonal input resistance in control (black) diazepam (green) and GABA-A antagonists picrotoxin and gabazine (red) conditions. (G) Time course of diazepam bath perfusion followed by GABA-A antagonist bath perfusion on the normalized axonal input resistance. Hexamethonium chloride (200 μM), sulpiride (1 μM), atropine (30 nM), and CGP 55845 (200 nM) were used to block nAChRs, D2Rs, mAChRs, and GABA-BRs respectively.

The online version of this article includes the following source data and figure supplement(s) for figure 6:

**Source data 1.** Raw values used for plots in *Figure 6* and *Figure 6—figure supplement 1*.
**Figure supplement 1.** Inhibition of GABA-A alone does not significantly enhance dopamine release.

dopamine release in the presence of cholinergic interneuron-mediated transmission, but also GABA-A receptors located directly on dopamine neuron axons to inhibit dopamine release.

To understand the mechanism behind this inhibition in the dopamine neuron axons we performed direct recordings from the axon. First, we sought to investigate whether these axonal GABA-A receptors are directly modulated by diazepam. For this experiment we puffed on muscimol in a voltage-clamp recording of the main axon, and then bath perfused diazepam (10 μM). Diazepam dramatically increased the amplitude of the muscimol-evoked current (*Figure 6C,E*). We also tested the effect of diazepam on the input resistance by giving a small voltage step (*Figure 6C*). We found

that, in the main axon, there was no effect of diazepam perfusion on the axonal input resistance (*Figure 6D*). This set of experiments shows that diazepam directly targets axonal GABA-A receptors on dopamine neurons, but in the medial fiber bundle GABA-A agonists must be exogenously applied to observe the effects of the drug.

Given the effect of diazepam on dopamine release reported above, we hypothesized that diazepam might be acting in concert with the striatal GABA tone to modulate dopamine neuron axons. When we recorded from striatal dopamine neuron axons and bath perfused diazepam, we found that diazepam decreased the input resistance of the axon in the striatum by an average of 22.7 ± 6.2% (*Figure 6F,G*), without any additional application of a GABA agonist. These results indicate a tonic level of GABA-A receptor activity that is enhanced by the application of diazepam, which decreases the input resistance of dopamine neuron axons and potentiates shunting inhibition. Thus, our data provide direct evidence that diazepam targets axonal GABA-A receptors to inhibit the release of striatal dopamine.

## Discussion

Here we examine the influence of GABA-A receptors on the excitability of dopaminergic neuron axons and the release of dopamine in the dorsal striatum. To test this, we performed whole-cell and perforated-patch recordings from the main axon located within the medial forebrain bundle as well as in the branched, signaling axon located in the striatum. Using this approach, we provide direct evidence that GABA-A receptors are present on the axons of midbrain dopaminergic neurons. We show that these receptors modulate propagation of action potentials in the axon through a combination of sodium channel inactivation and shunting inhibition. Finally, we demonstrate that diazepam (Valium), a commonly prescribed broad-spectrum benzodiazepine, enhances axonal GABA-A receptors, resulting in shunting and subsequent inhibition of dopamine release. Together, these experiments reveal the mechanisms of GABA-A receptor modulation of dopamine release and provide new insight into the role of axonal GABA-A receptors in the actions of benzodiazepines in the striatum.

### Action potential firing in midbrain dopaminergic neuron axons

The shape of the axonal action potential and the pre-spike membrane potential are critical determinants of neurotransmitter release (*Augustine, 1990*; *Awatramani et al., 2005*; *Geiger and Jonas, 2000*; *Rowan et al., 2016*; *Sabatini and Regehr, 1997*). Our data show that these features of axonal action potentials differ substantially from those that have been classically associated with somatic firing in dopaminergic neurons (*Grace and Bunney, 1983*; *Ungless and Grace, 2012*). For example, action potentials in the soma of dopamine neurons are typically broad, while we find that axonal action potentials are narrow with an average halfwidth of 0.89 ms, in agreement with studies that have reported brief presynaptic action potentials in other neuronal cell types (*Alle and Geiger, 2006*; *Geiger and Jonas, 2000*; *Hallermann et al., 2012*; *Kole et al., 2007*). We also find that action potentials are initiated from spike thresholds that are 14.3 mV more hyperpolarized than somatic spikes. Furthermore, the average non-spike voltage recorded in both the main axon and striatal axon is 6.7 mV and 8.8 mV more negative than values reported for the soma, respectively, which also fits with data from cortical layer five pyramidal neurons (*Hu and Bean, 2018*).

The hyperpolarized axonal interspike potential has possible functional implications on the control of dopamine release. First, the hyperpolarized axonal interspike voltage would likely maximize the availability of low-threshold channels such as L- and T-type calcium channels in axons, both of which are known to couple to dopamine release in the dorsal striatum (*Brimblecombe et al., 2015*). Second, activation of somatodendritic dopamine D2-receptors typically results in membrane hyperpolarization which then raises the question of how these receptors control axonal excitability and transmitter release. In the soma, D2-receptors have been shown to inhibit firing through activation of G-protein activated inwardly rectifying (GIRK2) potassium channels (*Beckstead et al., 2004*) and inhibition of the sodium leak channel NALCN (*Philippart and Khaliq, 2018*). In axons however, D2-receptors are thought to activate Kv1 channels (*Martel et al., 2011*). The hyperpolarized membrane potential of the axon suggests that further hyperpolarization by Kv1 may be limited by the potassium reversal potential and may not be the main mechanism of dopamine inhibition. Rather, Kv1-

mediated shunting and/or changes in spike shape are likely to contribute to the D2-dependent inhibition of dopamine release.

In somatic recordings of pacemaking, dopaminergic neurons exhibit a gradual depolarization of the interspike voltage thought to be critical for the generation of spontaneous activity (*Kang and Kitai, 1993*; *Khaliq and Bean, 2008*). By contrast, our data from distal recordings show that the slope of the interspike axonal membrane potential was shallow. The shallower interspike depolarization in the axon suggests that pacemaking in dopaminergic neurons results largely from the intrinsic excitability of the soma and dendrites. Furthermore, the hyperpolarized axonal threshold potential suggests that our recording site in the axon is distal to the site of action potential initiation, which is the axon initial segment (*Häusser et al., 1995*; *Shu et al., 2007*). Therefore, these observations argue against the axon as a third site of oscillation generation after the soma and dendrites (*Pissadaki and Bolam, 2013*). It is important to note that although the mixture of conductances present in axons does not favor spontaneous activity, it is still possible that the conductances that drive somatic depolarization such as NALCN and HCN may also be present in axons. In fact, a recent study found a positive correlation between the length of the axon initial segment and the spontaneous firing rate, suggesting that the conductances present in the axon initial segment speed firing (*López-Jury et al., 2018*; *Meza et al., 2018*). However, a different study found that the geometry of the axon initial segment negligibly affects the firing rate (*Moubarak et al., 2019*). In addition to the axon initial segment geometry, future work should focus on determining the axonal conductances that enable and control firing rate and spike transmission.

## Axonal GABA-A receptors on dopaminergic neuron axons are depolarizing

The published literature has shown that large differences exist in the reversal potential of axonal chloride-based conductances when comparing between neuronal cell types. For example, a careful study of the GABA reversal potential in the axon initial segment of cortical layer 2/3 pyramidal neurons demonstrated that $E_{GABA}$ shifts from depolarizing to hyperpolarizing with age, eventually settling at negative values near the somatic resting potential in adult mice (~ −87 mV, *Rinetti-Vargas et al., 2017*). Similarly, hyperpolarized GABA reversal potential values were reported from proximal axons of layer five pyramidal neurons from rats (*Xia et al., 2014*). By contrast, axonal GABA-A receptors on the mossy fiber bouton (*Ruiz et al., 2010*), cultured Purkinje neuron terminals (*Zorrilla de San Martin et al., 2017*) as well as axonal glycine receptors on the calyx of Held (*Price and Trussell, 2006*) have reported axonal chloride-based conductances that are depolarizing relative to resting membrane potential.

In this study, we demonstrate that the average reversal potential of GABA-mediated currents in dopamine neuron axons, when considered relative to the average axonal interspike membrane potential of −68 mV, is also depolarized at −56 mV. Our recordings were performed in adult mice (ages 6–17 weeks, median of 15.5 weeks) suggesting that the depolarized reversal potential that we obtained represents the value in mature axons. Interestingly, the reversal potential for somatodendritic GABA currents in dopaminergic neurons is also depolarized at −63 mV due to relatively low expression of the K-Cl cotransporter KCC2 (*Gulácsi et al., 2003*), which is similar to the average interspike membrane potential of dopamine neurons during pacemaking. Therefore, activation of somatodendritic GABA-A receptors reduces spiking primarily through shunting with relatively little change in the membrane potential.

## Mechanism of axonal GABA-A receptor mediated inhibition of striatal dopamine release

Despite the depolarized GABA reversal potential in distal axons, our findings show that activation of axonal GABA-A receptors results in inhibition of dopamine release. Although this is consistent with work from spinal cord (*Curtis and Lodge, 1982*; *Eccles et al., 1961*; for a review see, *Trigo et al., 2008*), these results differ from previous studies that have found axonal GABA-A receptors enhance synaptic transmission in cerebellar parallel fibers (*Dellal et al., 2012*; *Howell and Pugh, 2016*; *Khatri et al., 2019*; *Pugh and Jahr, 2011*) hippocampal mossy fibers (*Ruiz et al., 2010*), terminals of cerebellar Purkinje neurons (*Zorrilla de San Martin et al., 2017*) and in layer 2/3 pyramidal neurons of the cortex (*Szabadics et al., 2006*).

What features distinguish dopaminergic neuron axons, and contribute to the inhibitory effect of GABA-A receptors on transmitter release? The answer to this question is currently unknown. However, one possibility is that dopaminergic neurons differ dramatically from these other cell types in axonal architecture. For example, parallel fibers and mossy fibers are unbranching, or branch only infrequently. On the other hand, dopamine neuron axons are among the most branching processes in the brain, forming new bifurcations an average of 31 μm, and possessing an average total length of 467,000 μm (*Matsuda et al., 2009*), from which we can estimate roughly 15,000 total branches per cell. Our data suggests that this unusually high frequency of branching may lead to stronger attenuation of propagating spikes. We found that activation of GABA-A receptors had only subtle effects on axonal calcium signals at proximal imaging sites while axonal calcium signals at distal imaging sites were dramatically reduced. We also provide evidence that GABA-A activation reduces the height of axonal action potentials. Perhaps the added propagation distance enhances the effects GABA-A receptors in shortening action potentials and depressing dopamine release. More generally, the density of voltage-gated sodium channels and other channels that support active propagation are challenged by axonal GABA-A receptors, which may have a stronger effect in the thin, highly branching distal axon. In some cases, therefore, the shortening of axonal action potentials may eventually lead to an increase in branch point failures. Future experiments should seek to use whole-cell axonal recordings to directly measure action potential waveforms in the striatal axon following GABA-A activation.

Past studies have proposed that presynaptic GABA-A receptors exert their effects through either shunting inhibition or sodium channel inactivation (*Trigo et al., 2008*). Because of the lack of experimental access to the axonal compartment, however, direct tests of this hypothesis have previously been limited to large terminal structures. In the rat posterior pituitary nerve terminals, GABA was shown to produce large depolarizations that led to strong inactivation of sodium channels, while shunting was thought to play little role in inhibition of secretion from terminals (*Zhang and Jackson, 1993*). Here, we demonstrate in the thin, unmyelinated axons of dopaminergic neurons that shunting and depolarization-mediated inactivation of sodium channels contribute nearly equally to GABA-A receptor mediated alteration of action potential shape and the subsequent inhibition of striatal dopamine release. Under conditions of tonic GABA-A receptor activation, these two inhibitory mechanisms will be especially prominent, particularly in an electrically tight compartment like the axon where tiny fluctuations of GABA-A activity can cause large changes in membrane voltage and input resistance. Furthermore, we found that these two mechanisms of inhibition differentially affect action potential waveforms. While depolarization-mediated sodium channel inactivation both reduces spike height and slows the rate of action potential rise, shunting inhibition only affects spike height.

## Effect of benzodiazepines on axons

Benzodiazepines can have rewarding effects that, in some cases, can lead to habit formation (*Blanco et al., 2018*; *Tan et al., 2011*). The rewarding actions of benzodiazepines are thought to involve potentiation of GABA-A receptors located on inhibitory GABAergic neuron within the VTA which then results in disinhibition of VTA dopaminergic neurons (*Tan et al., 2010*). As is the case with other drugs of abuse that disinhibit dopamine neurons (e.g. opioids), benzodiazepines would be expected then to increase the somatic firing rate and subsequent dopamine release in the striatum. Instead, studies of awake behaving rats show that systemic diazepam administration increases the frequency of dopamine release events but decreases the amplitude of these release events (*Schelp et al., 2018*). The apparent disparity in these results can be reconciled by our observation that axonal GABA-A receptors on dopaminergic neuron axons are enhanced by diazepam. This enhancement of GABA-A receptors leads to a decrease in dopamine release through a combination of shunting inhibition and depolarization-mediated sodium channel inactivation. Therefore, we propose that the effects of drugs that pharmacologically target GABA-A receptors such as ethanol, barbiturates, and other sedatives should be reexamined considering their potential effects on axonal GABA-A receptors.

Of note, most of these compounds that target GABA-A receptors are not direct agonists, but rather positive allosteric modulators. Diazepam, for example, has been shown to both increase single channel conductance of GABA-A receptors (*Eghbali et al., 1997*) and increase receptor affinity for GABA (*Campo-Soria et al., 2006*). Therefore, diazepam does not induce substantial receptor

opening, but instead enhances the affinity and efficacy of orthosteric agonists, an effect that may depend on novel GABA-A receptor auxiliary subunits (*Han et al., 2019*). Our observation that diazepam potentiates GABA-A receptors located on dopaminergic axons consequently indicates a tonic level of activation of these receptors. GABA-A receptors are well established to exhibit tonic activity, as shown by a change in holding current and a reduction in the recording RMS noise while in voltage clamp (*Semyanov et al., 2004*). This tonic activity is also important for the mechanism of action of some drugs, such as the anticonvulsant Vigabatrin (*Overstreet and Westbrook, 2001*), showing both physiological and pathological roles for tonic activation of GABA-A receptors in cellular physiology.

Past studies have observed tonic GABA currents on neurons in the striatum. For example, dopamine D2 receptor-expressing medium spiny neurons have been shown to exhibit GABA-A receptor-mediated tonic currents (*Ade et al., 2008*). The tonic currents in this study were sensitive to the sodium channel blocker, tetrodotoxin, suggesting that the GABA tone originates from spontaneous synaptic release. Other work suggests that a portion of the tonic GABA in striatum is released through action potential-independent processes (*Wójtowicz et al., 2013*). Regarding control of striatal dopamine, recent studies suggest that tonic GABA inhibits dopamine release primarily through activation of GABA-B receptors, with little to no contribution of GABA-A receptors in this process (*Brodnik et al., 2019*; *Lopes et al., 2019*). Our results also show that blockade of GABA-A receptors with gabazine has little effect on dopamine release under control conditions. Following application of diazepam, however, we observe a significant reduction in input resistance of dopaminergic neuron axons. Importantly, these experiments were performed in the absence of electrical or optical stimulation which argues strongly for the involvement of GABA tone in the diazepam-mediated inhibition of striatal dopamine. Together, these findings suggest that the GABA-A receptors on dopamine neuron axons are not apposed to vesicular release sites but are likely spread along the axon shaft and boutons where they are activated by extrasynaptic GABA. Therefore, modulators of GABA-A receptors such as benzodiazepines will likely function globally to influence the shape of action potentials and affect propagation by driving subthreshold depolarization and reducing axonal input resistance across the axonal arbor.

We were surprised to find that the effect of diazepam on dopamine release was strongest when cholinergic transmission was left intact. Although the precise explanation for this unknown, it is likely that the lower axonal input resistance in the presence of diazepam results in reduced integration of cholinergic input. In addition, cholinergic interneurons also have been shown to express benzodiazepine-sensitive GABA-A receptors (*Yan and Surmeier, 1997*) and it would be interesting to determine whether these also include axonal GABA-A receptors. Lastly, cholinergic interneurons innervate multiple cell types throughout the striatum which may contribute to tonic GABA and thus may control tonic dopamine. Future studies should explore additional circuit elements that may contribute to GABAergic control of striatal dopamine release, particularly in the case of axons that participate in direct axo-axonal communication.

In sum, this report shows direct evidence for GABA-A receptors on dopamine neuron axons. These receptors act mechanistically in the axon through a combination of increased shunting inhibition and sodium channel inactivation. Functionally, this results in reduced action potential propagation through the axonal arbor and decreased dopamine release, especially distal to the site of action potential initiation. Finally, benzodiazepines act directly on axonal GABA-A receptors to enhance the effects of GABA tone in the striatum, making the axons leakier and potentially weakening subthreshold integration.

## Materials and methods

**Key resources table**

| Reagent type (species) or resource | Designation | Source or reference | Identifiers | Additional information |
|---|---|---|---|---|
| Genetic reagent (adeno-associated virus) | AAV1-hsyn-FLEX-CoChR-GFP | UNC vector core | Boyden, E. | |

*Continued on next page*

*Continued*

| Reagent type (species) or resource | Designation | Source or reference | Identifiers | Additional information |
|---|---|---|---|---|
| Genetic reagent (adeno-associated virus) | AAV9-CAG-FLEX-TdTomato | Penn vector core | | |
| Genetic reagent (adeno-associated virus) | AAV9-Syn-FLEX-jGCaMP7f | Janelia | | |
| Genetic reagent (adeno-associated virus) | AAV9-hSyn-dLight1.2 | Lin Tian | | |
| Chemical compound, drug | Streptavidin cy5 conjugate | Invitrogen | SA1011 | 1:1000 |
| Chemical compound, drug | SR95531 hydrobromide (Gabazine) | Tocris | Cat# 1262 | |
| Chemical compound, drug | SR95531 hydrobromide (Gabazine) | Hello Bio | Cat# HB0901 | used only in *Figure 6—figure supplement 1A and B* |
| Chemical compound, drug | CGP55845 hydrochloride | Tocris | Cat# 1248 | |
| Chemical compound, drug | D-AP5 | Tocris | Cat# 0106 | |
| Chemical compound, drug | Atropine | MilliporeSigma | Cat# A0132 | |
| Chemical compound, drug | NBQX disodium salt | Tocris | Cat# 1044 | |
| Chemical compound, drug | (±)-Sulpiride | MilliporeSigma | Cat# S8010 | |
| Chemical compound, drug | Tetrodotoxin (TTX) | Tocris | Cat# 1078 | |
| Chemical compound, drug | $\lambda$-Aminobutyric Acid (GABA) | MilliporeSigma | Cat# A5835 | |
| Chemical compound, drug | Picrotoxin | MilliporeSigma | Cat# P1675 | |
| Chemical compound, drug | Hexamethonium Chloride | MilliporeSigma | Cat# H2138 | |
| Chemical compound, drug | Diazepam | MilliporeSigma | Cat# D0899 | |
| Chemical compound, drug | Gramicidin | MilliporeSigma | Cat# G5002 | |
| Chemical compound, drug | Muscimol | Tocris | Cat# 0289 | |
| Chemical compound, drug | Gelatin from cold water fish skin | MilliporeSigma | Cat# G7041 | |
| Chemical compound, drug | (+)-Sodium L-ascorbate | MilliporeSigma | Cat# A4034 | |
| Chemical compound, drug | Sodium Pyruvate | MilliporeSigma | Cat# P5280 | |
| Chemical compound, drug | Thiourea | MilliporeSigma | Cat# T7875 | |
| Chemical compound, drug | Triethanolamine | MilliporeSigma | Cat# 90279 | |
| Chemical compound, drug | N,N,N',N'-Tetrakis-(2-hydroxypropyl) ethylenediamine | TCI | Cat# T0781 | |
| Chemical compound, drug | Urea | MilliporeSigma | Cat# U5128 | |

*Continued on next page*

*Continued*

| Reagent type (species) or resource | Designation | Source or reference | Identifiers | Additional information |
|---|---|---|---|---|
| Strain, strain background (*Mus musculus*) | Ai95-RCL-GCaMP6f-D: Cg-Gt(ROSA)26Sor(tm95.1 (CAG-GCaMP6f)Hze)/MwarJ | The Jackson Laboratory | Cat# 028865 | male and female over 6 weeks of age |
| Strain, strain background (*Mus musculus*) | Ai9: Gt(ROSA)26Sor (tm9(CAG-tdTomato)Hze) | The Jackson Laboratory | Cat# 007909 | male and female over 6 weeks of age |
| Strain, strain background (*Mus musculus*) | TH-GFP (Tg(TH-EGFP)1Gsat | NIH MMRRC | | male and female over 6 weeks of age |
| Strain, strain background (*Mus musculus*) | C57/Bl6J Wild Type | The Jackson Laboratory | Cat# 000664 | male and female over 6 weeks of age |
| Strain, strain background (*Mus musculus*) | B6.SJL-Slc6a3 (tm1.1(cre)Bkmn/ J (DAT-IRES-cre) | The Jackson Laboratory | Cat# 006660 | male and female over 6 weeks of age |
| Strain, strain background (*Mus musculus*) | Ai32: B6.Cg – Gt(ROSA) 26Sor(tm32(CAG-COP4*H 143R/EYFP)Hze) | The Jackson Laboratory | Cat#024109 | male and female over 6 weeks of age |
| Software, algorithm | Igor Pro 6 | Wavemetrics | RRID:SCR_000325 | |
| Software, algorithm | FIJI | PMID:22743772 | RRID:SCR_002285 http://fiji.sc | |
| Software, algorithm | Prism 8 | GraphPad | RRID:SCR_002798 | |
| Software, algorithm | pClamp 11 | Axon Instruments | RRID:SCR:011323 | |

## Experimental model and subject details

All animal handling and procedures were approved by the animal care and use committee (ACUC) for the National Institute of Neurological Disorders and Stroke (NINDS) at the National Institutes of Health (protocol 1322). Mice of both sexes were used throughout the study. Mice that underwent viral injections were injected at postnatal day 18 or older and were used for ex vivo electrophysiology and imaging 3–12 weeks after injection. The following strains were used: DAT-Cre (SJL-Slc6a3 (tm1.1(cre)Bkmn/J, The Jackson Laboratory Cat#006660); Ai95-RCL-GCaMP6f-D (Cg-Gt(ROSA)26Sor (tm95.1(CAG-GCaMP6f)Hze)/MwarJ, The Jackson Laboratory Cat#028865); Ai9 (Gt(ROSA)26Sor(tm9 (CAG-tdTomato)Hze), The Jackson Laboratory Cat#007909); TH-GFP (Tg(TH-EGFP)1Gsat) NIH MMRRC; C57/Bl6J Wild Type, The Jackson Laboratory Cat#000664; Ai32 (B6.Cg – Gt(ROSA)26Sor (tm32(CAG-COP4*H143R/EYFP)Hze), The Jackson Laboratory, Cat#024109).

## Method details

### Viral injections

All stereotaxic injections were conducted on a Stoelting QSI (Cat#53311). Mice were maintained under anesthesia for the duration of the injection and allowed to recover from anesthesia on a warmed pad. The AAV9-CAG-FLEX-TdTomato (Penn Vector Core), AAV-Syn-FLEX-jGCaMP7f (*Dana et al., 2019*), and AAV9-hSyn-dLight1.2 (*Patriarchi et al., 2018*) viruses (0.5–1 µl) were injected bilaterally into either the medial dorsal striatum (X:±1.7 Y: +0.8 Z: −3.3) or the SNc (X:±1.9 Y: −0.5 Z: −3.9) via a Hamilton syringe. At the end of the injection, the needle was raised at a rate of 0.1 to 0.2 mm per minute for 10 minutes before the needle was removed.

## Slicing and electrophysiology

Brain slice experiments were performed on male and female adult mice of at least 6 weeks in age. Mice were anesthetized with isoflurane, decapitated, and brains rapidly extracted. Horizontal sections (electrophysiology, dLight, calcium imaging) or coronal sections (voltammetry) were cut at 330–400 µm thickness on a vibratome while immersed in warmed, modified, slicing ACSF containing (in mM) 198 glycerol, 2.5 KCl, 1.2 $NaH_2PO_4$, 20 HEPES, 25 $NaHCO_3$,10 glucose, 10 $MgCl_2$, 0.5 $CaCl_2$, 5 Na-ascorbate, 3 Na-pyruvate, and two thiourea. Cut sections were promptly removed from the slicing chamber and incubated for 30–60 min in a heated (34°C) chamber with holding solution containing (in mM) 92 NaCl, 30 $NaHCO_3$, 1.2 $NaH_2PO_4$, 2.5 KCl, 35 glucose, 20 HEPES, 2 $MgCl_2$, 2 $CaCl_2$, 5 Na-ascorbate, 3 Na-pyruvate, and 2 thiourea. Slices were then stored at room temperature and used

30 minutes to 6 hours later. Following incubation, slices were moved to a heated (33–35˚C) recording chamber that was continuously perfused with recording ACSF (in mM): 125 NaCl, 25 NaHCO$_3$, 1.25 NaH$_2$PO$_4$, 3.5 KCl, 10 glucose, 1 MgCl$_2$, 2 CaCl$_2$. Whole-cell recordings were made using borosilicate pipettes (5–10 MΩ, axon; 2–3 MΩ, soma) filled with internal solution containing (in mM) 122 KMeSO$_3$, 9 NaCl, 1.8 MgCl$_2$, 4 Mg-ATP, 0.3 Na-GTP, 14 phosphocreatine, 9 HEPES, 0.45 EGTA, 0.09 CaCl$_2$, adjusted to a pH value of 7.35 with KOH. For high chloride experiments, KMeSO$_3$ was substituted with KCl.

Perforated-patch recordings were made using borosilicate pipettes (5–10 MΩ) filled with internal solution containing (in mM) 135 KCl, 10 NaCl, 2 MgCl$_2$, 10 HEPES, 0.5 EGTA, 0.1 CaCl$_2$, adjusted to a pH value of 7.43 with KOH, 278 mOsm. Pipette tips were back-filled first with ~1 µL of internal lacking gramicidin followed by internal containing either 4–8 (perforated-patch on the main axon) or 80–100 µg/mL (perforated-patch in the striatum) gramicidin. Patch integrity was monitored by the addition of Alexa-488 to the gramicidin-containing internal. Experiments were discarded upon visual evidence of membrane rupture (Alexa-488 entering the axon).

To enable post-hoc reconstruction, pipette solutions in some experiments included 0.1–0.3% w/v neurobiotin (Vector Labs), and 0.01 mM AlexaFluor 594 hydrazide or AlexaFluor 488 hydrazide. Current clamp recordings were manually bridge balanced. Liquid junction potential for KMeSO$_3$ based internal solutions was −8 mV and was corrected offline.

Electrical stimulation was evoked with tungsten bipolar electrodes (150 µm tip separation, MicroProbes). For experiments where the site of electrical stimulation is distal to the site of imaging or recording, electrodes were placed at the caudal end of horizontal brain slices, or at the medial end of coronal slices. Stimulations were evoked using an Isoflex (A.M.P.I.), amplitudes ranging from 0.1 to 75 V.

Pressure ejection was performed using a borosilicate micropipette pulled on a horizontal puller (pipette size ~2–4 MΩ). The pharmacological agent being tested, either GABA or muscimol, was added to a modified external solution containing (in mM): 125 NaCl, 25 NaHCO$_3$, 1.25 NaH$_2$PO$_4$, 3.5 KCl, 10 HEPES, 0.01 either Alexa 488 (for experiments in DAT-Cre x Ai9 animals) or Alexa 594 (for experiments in TH-GFP animals), final osmolarity 280–290 mOsm. This puffing solution was then spin filtered, loaded into the glass pipette, and lowered to within 30–50 µm of the axon using a micro-manipulator. The puffing solution was then applied onto the axon with a short pressure ejection (80–300 ms in duration) using a PV 820 Pneumatic PicoPump (WPI).

## Fast-scan cyclic voltammetry (FSCV)

For all voltammetry experiments the method is as follows. Cylindrical carbon-fiber electrodes (CFEs) were prepared with T650 fibers (6 µm diameter,~150 µm of exposed fiber) inserted into a glass pipette and filled with KCl (3 M). Before use, the CFEs were conditioned with 8 ms long triangular voltage ramp (−0.4 to +1.2 and back to −0.4 V versus Ag/AgCl reference at 4 V/s) delivered every 15 ms. CFEs showing current above 1.8 µA or below 1.0 µA in response to the voltage ramp around 0.6 V were discarded. A triangular voltage ramp was passed through the fiber from −400 mV to 1200 mV, and returned to −400 mV. The ramp was run at a rate of 400 V/s, every 100 ms.

For voltammetry experiments performed by E.L.T., the method is as follows. Dopamine transients were evoked by a 2 ms 470 nm LED (ThorLabs) pulse every 60 seconds. Selective channel rhodopsin stimulation of dopamine neuron axons was achieved by injecting cre-dependent CoChR channel rhodopsin into the SNc of *DAT-cre* transgenic animals. Peak dopamine currents were calculated from voltammograms created in Igor Pro (Wavemetrics).

For voltammetry experiments performed by J.H.S., the method is as follows. Mice were anesthetized with isoflurane and sacrificed by decapitation. Brains were sliced in sagittal orientation at 240 µm thickness with a vibratome (VT-1200S Leica) in an ice-cold cutting solution containing (in mM) 225 sucrose, 13.9 NaCl, 26.2 NaHCO$_3$, 1 NaH$_2$PO$_4$, 1.25 glucose, 2.5 KCl, 0.1 CaCl$_2$, 4.9 MgCl$_2$, and three kynurenic acid. Slices were incubated for 20 min at 33˚C in artificial cerebrospinal fluid (ACSF) containing (in mM) 124 NaCl, 1 NaH$_2$PO$_4$, 2.5 KCl, 1.3 MgCl$_2$, 2.5 CaCl$_2$, 20 glucose, 26.2 NaHCO$_3$, 0.4 ascorbic acid, and maintained at room temperature prior recordings. Slices were placed in a submerged chamber perfused at 2 ml/min with ACSF at 32˚C using an in-line heater (Harvard Apparatus).

DA transients were evoked by brief light pulse (0.6–0.8 ms) through an optical fiber (200 µm/0.22 NA) connected to a 470 nm LED (2 mW; ThorLabs) delivered every 2 min. Data were collected with

a modified electrochemical headstage (CB-7B/EC retrofit with 5 MΩ resistor) using a Multiclamp 700B amplifier (Molecular Devices) after being low-pass filtered at 10 kHz and digitized at 100 kHz using custom-written software in Igor Pro named VIGOR (*Bock and Shin, 2018*; *Bock, 2018*) running mafPC software (*Xu-Friedman, 2019*). Decay time constants were obtained with a single exponential fit of the derivative of the falling phase of DA transient curve.

### Fluorescent imaging

Calcium was measured in dopamine neuron axons of the medial dorsal striatum using the GCaMP6f mouse bred with the DAT-Cre mouse, or with viral injection of CRE-dependent jGCaMP7f into the SNc of DAT-cre mice. These data were combined in the results. All calcium imaging experiments were performed in the presence of atropine (30 nM), sulpiride (1 μM), hexamethonium chloride (200 μM), and CGP55845 (200 nM). A white light LED (Thorlabs; SOLIS-3C) was used in combination with a GFP filter set. A photodiode (New Focus) was mounted on the top port of the Olympus BX-51WI microscope.

### Immunohistochemistry, clearing, confocal imaging, and neural reconstructions

After electrophysiology or imaging, slices were fixed overnight in 4% paraformaldehyde (PFA) diluted in phosphate buffer (PB, 0.1M, pH 7.6). Slices were subsequently stored in PB until immunostaining and cleared using a modified CUBIC protocol, chosen because it does not quench endogenous fluorescence (*Susaki et al., 2015*). For the immunostaining/CUBIC clearing, all steps were performed at room temperature on a shaker plate. Slices were placed in CUBIC reagent 1 for 1–2 days, washed in PB 3 × 1 hour each, placed in blocking solution (0.5% fish gelatin (MilliporeSigma) in PB) for 3 hours. Slices were directly placed in streptavidin-Cy5 conjugate at a concentration of 1:1000 in PB for 2–3 days. Slices were washed 3 times for 2 hr each and were then placed in CUBIC reagent two overnight. Slices were mounted on slides in reagent two in frame-seal incubation chambers (Bio-Rad SLF0601) and coverslipped (#2 glass). Slices were imaged through 20×, 0.8 nA and 5×, 0.3 nA objectives on an LSM 800 confocal microscope (Zeiss), and taken as tiled z-stacks using Zen Blue software in the NINDS light imaging facility.

Main axons were reconstructed and measured using Simple Neurite Tracer in FIJI (*Longair et al., 2011*). Of axons with a positively identified soma, the majority were found in the substantia nigra *pars compacta*, with some found in the ventral tegmental area. Striatal axons were reconstructed using Neurolucida (MBF bioscience).

### Drugs

All salts and all drugs not otherwise stated were from MilliporeSigma. Fluo5F and Alexa594 (Life Technologies), gabazine, d-AP5, hexamethonium chloride, oxotremorine M, GABA, and muscimol, were dissolved in deionized water. Sulpiride, quinpirole, picrotoxin, CGP55845 (Tocris), NBQX, and diazepam were dissolved in DMSO. Atropine was dissolved in DMSO and then diluted 1:10 in deionized water.

### Quantification and statistical analysis

Analysis was conducted in Igor Pro and Prism 8 (GraphPad). Data in text are reported as mean ($\bar{x}$) ± SEM for parametric or median ($\tilde{x}$) for non-parametric data. Error bars on graphs are indicated as ± SEM. Box plots show medians, 25 and 75% (boxes) percentiles, and 10 and 90% (whiskers) percentiles. For parametric data, t-tests were used for two-group comparison, and ANOVA tests were used for more than two group comparisons, followed by a Bonferonni or Šidák post-hoc test for analysis of multiple comparisons. For non-parametric data sets, Mann-Whitney U tests were used to compare two groups while the Kruskal-Wallis test was used to compare more than two groups. For linear regression analysis, the Straight Line analysis function was used in Prism, and an extra sum-of-squares F test was performed to determine significant differences in slope between data sets on the same plot, and to determine whether a line or exponential decay model fits the data better. For exponential fits, the One Phase Decay analysis function in Prism was used to fit a standard curve.

## Acknowledgements

We would like to thank Dr. Veronica Alvarez (NIAAA) for input on experiments and sharing lab equipment for voltammetry experiments, and to Dr. Carolyn Smith and the NINDS Light Imaging Facility for the training and equipment to take confocal images of cleared tissue. Finally, we would like to acknowledge the Khaliq lab for their input on experiments, data presentation and the text. Funding for this research was provided by an NINDS intramural research program grant NS003134 to ZMK and a Center for Compulsive Behaviors fellowship, Intramural Research Program, NIH, awarded to PFK.

## Additional information

### Funding

| Funder | Grant reference number | Author |
| --- | --- | --- |
| National Institutes of Health | NS003134 | Zayd M Khaliq |
| National Institutes of Health | ZIA-AA000421 | Jung Hoon Shin |
| National Institutes of Health | | Paul F Kramer |

The funders had no role in study design, data collection and interpretation, or the decision to submit the work for publication.

### Author contributions

Paul F Kramer, Conceptualization, Resources, Data curation, Software, Formal analysis, Validation, Investigation, Visualization, Methodology, Writing - original draft, Writing - review and editing; Emily L Twedell, Jung Hoon Shin, Formal analysis, Investigation, Methodology, Writing - review and editing; Renshu Zhang, Resources, Methodology; Zayd M Khaliq, Conceptualization, Resources, Supervision, Funding acquisition, Methodology, Writing - original draft, Project administration, Writing - review and editing

### Author ORCIDs

Paul F Kramer (iD) https://orcid.org/0000-0002-0095-3712
Jung Hoon Shin (iD) http://orcid.org/0000-0002-9892-1275
Zayd M Khaliq (iD) https://orcid.org/0000-0002-1445-1457

### Ethics

Animal experimentation: All animal handling and were approved by the animal care and use committee (ACUC) for the National Institute of Neurological Disorders and Stroke (NINDS) at the National Institutes of Health under Animal Study Proposal (ASP) #1322. All surgery was performed under iso-flurane anesthesia, followed by close monitoring and treatment. Every effort was made to minimize suffering.

### Decision letter and Author response

Decision letter https://doi.org/10.7554/eLife.55729.sa1
Author response https://doi.org/10.7554/eLife.55729.sa2

## Additional files

### Supplementary files

• Transparent reporting form

### Data availability

Data files containing data shown in Figures 1-6 have been provided.

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
