## [Decision Letter]

**Acceptance summary:**

Axonal receptors on dopamine neurons allow local regulation of dopamine release at a distance from the cell body. The authors here use elegant electrophysiological and imaging approaches to show that GABAA receptors depolarize distal dopaminergic axons to reduce axonal excitability, providing new understanding of how GABA inhibits dopamine release in the striatum.

**Decision letter after peer review:**

Thank you for submitting your article "Axonal mechanisms mediating GABA-A receptor inhibition of striatal dopamine release" for consideration by *eLife*. Your article has been reviewed by three peer reviewers, one of whom is a member of our Board of Reviewing Editors, and the evaluation has been overseen by John Huguenard as the Senior Editor. The following individual involved in review of your submission has agreed to reveal their identity: Christopher Ford (Reviewer #2).

The reviewers have discussed the reviews with one another and the Reviewing Editor has drafted this decision to help you prepare a revised submission.

Summary:

Here the authors examine how GABAA receptors expressed on the axons of dopamine neurons inhibit release of dopamine, providing a mechanistic explanation for prior findings that GABA inhibits DA release in the striatum. Through a series of technically challenging electrophysiological and imaging experiments, the authors conclude that axonal GABAA receptors depolarize axons to reduce action potential propagation (spike height) via a combination of shunting and inactivation of sodium channels. They also conclude that the benzodiazepine diazepam acts on the endogenous tone of GABA present in the striatum to decrease DA release via shunting inhibition. Overall the conclusions are convincing and provide new insight into regulation of DA release as well as general mechanisms of axonal modulation by presynaptic receptors.

Essential revisions:

1) While the results are convincing that axonal GABAA receptors are present and can modulate axonal excitability, it is less clear that there is a tonic GABAA-mediated current due to ambient GABA in the slice. Analysis of effects of GABAA receptor antagonists alone should be expanded to illustrate the actions on endogenous GABA (i.e. as in Figures 3, 4 or 5). Currently the strongest data supporting endogenous GABAA receptor activation (Figure 5A) is complicated by co-application of CGP55845, which is an inverse agonist at GABAB receptors that are generally higher affinity than GABAA receptors. It is possible that ambient GABA is too low to significantly activate GABAA receptors but diazepam enables activation by increasing GABAA receptor affinity.

2) The authors suggest reliability of spike propagation in the highly branched distal axon may be reduced by GABA (subsection “Stronger effect of GABA-A inhibition on APs that undergo propagation”), but the calcium imaging results do not really support this conclusion (Figure 3D-F). There are a number of explanations that could account for the greater decrease in calcium signal in the distal axon, the simplest being that GABAAR expression increases in the distal axon (or Ecl is more depolarized). Also, the authors report a decrease in spike amplitude (Figure 4), but do not mention seeing greater spike failures when applying GABA, suggesting branch point failures are not common. Other mechanisms should be considered and discussed.

---

## [Author Response]

Essential revisions:1) While the results are convincing that axonal GABAA receptors are present and can modulate axonal excitability, it is less clear that there is a tonic GABAA-mediated current due to ambient GABA in the slice. Analysis of effects of GABAA receptor antagonists alone should be expanded to illustrate the actions on endogenous GABA (i.e. as in Figures 3, 4 or 5). Currently the strongest data supporting endogenous GABAA receptor activation (Figure 5A) is complicated by co-application of CGP55845, which is an inverse agonist at GABAB receptors that are generally higher affinity than GABAA receptors. It is possible that ambient GABA is too low to significantly activate GABAA receptors but diazepam enables activation by increasing GABAA receptor affinity.

This reviewer comment raises the question of whether tonic GABA is present in the slice. We will first respond to this by stating that our experiments testing direct axonal input resistance argue strongly for the presence of tonic GABA. In particular, we found that application of diazepam in the absence of any exogenously applied GABA application or synaptic stimulation resulted in a clear decrease in the input resistance of dopaminergic neuron axons in the striatum. The effect of diazepam on axonal input resistance was reversed by GABA-A receptor antagonists suggesting that the effect is not due to off-target effects but rather is specific to GABA-A receptors. In addition, we found that diazepam had no effect on the main axon demonstrating that tonic GABA-A mediated current is confined to dopaminergic neuron axons in striatum. Together, we believe that these experiments demonstrate that diazepam directly inhibits dopamine and provides strong evidence for the presence of tonic GABA in the striatum.

As suggested by the reviewers, it is unclear whether the tonic level of GABA-A receptor activity in striatum is strong enough to inhibit dopamine release in the absence of GABA-A positive modulators. To address this question, we performed new imaging experiments. We tested the effect of the GABA-A antagonist, gabazine, on stimulated dopamine release using dLight and found that gabazine alone does not significantly increase dopamine release. Therefore, although the concentration of tonic GABA is sufficient to inhibit dopamine release through activation of GABA-B receptors (Figure 6—figure supplement 1A, B), it is not sufficient to measurably inhibit dopamine release through GABA-A receptors under control conditions (i.e. in the absence of benzodiazepines). These new experimental data have been included in Figure 6—figure supplement 1C, D and a description of this experiment has been added to the subsection “Benzodiazepines enhance tonic GABA activity on dopamine neuron axons” of the Results section. It is important to note that our experiments were performed in deafferented brain slices that lack ongoing firing activity. However, in vivo experiments in other brain regions such as cerebellum have observed tonic GABA-A mediated inhibitory currents which significantly dampen action potential firing activity of granule neurons (Chadderton et al., 2004). Likewise, striatal neurons are also more active under in vivo conditions and in behaving animals. Therefore, future work should examine whether tonic GABA-A mediated currents play a role in controlling striatal dopamine release under these conditions.

2) The authors suggest reliability of spike propagation in the highly branched distal axon may be reduced by GABA (subsection “Stronger effect of GABA-A inhibition on APs that undergo propagation”), but the calcium imaging results do not really support this conclusion (Figure 3D-F). There are a number of explanations that could account for the greater decrease in calcium signal in the distal axon, the simplest being that GABAAR expression increases in the distal axon (or Ecl is more depolarized).

The reviewers raised the point that the greater decrease in axonal calcium signals by muscimol observed in the rostral striatum (Figure 5A-C) may be due to regional differences in inhibitory axonal signaling such as stronger GABA-A expression levels and/or more depolarized chloride reversal potentials. To examine this possibility, we tested the effect of muscimol on axonal calcium signals that were both stimulated and imaged within the rostral striatum. Similar to analogous experiments performed in the caudal striatum (Figure 5D-F), we observed little effect of muscimol on axonal calcium signals that were both stimulated and imaged locally within rostral striatum (Figure 5G-I). Therefore, these new data demonstrate that muscimol has little effect on axonal calcium signals that result from local stimulation of either caudal or rostral striatum.

By contrast, GABA-A receptor activation results in substantial inhibition of axonal calcium signals imaged in rostral striatum when those signals were generated by distal stimulation of the caudal striatum. These observations demonstrate that calcium signals in axons of dopaminergic neurons within the striatum are most greatly affected by GABA-A receptor activation under conditions involving spike propagation over large distances.

Also, the authors report a decrease in spike amplitude (Figure 4), but do not mention seeing greater spike failures when applying GABA, suggesting branch point failures are not common. Other mechanisms should be considered and discussed.

To clarify this point, the experiments described in Figure 4 test the effects of GABA-A receptor activation on action potentials recorded in the main axon trunk (i.e. before the axon has branched). Secondly, the GABA puffer pipette was typically placed very closely to the recording site which means that propagation occurred only over very short distances in these experiments. Although we do not observe clear failures under these experimental conditions, however, we believe that the decrease in spike amplitude following GABA application to unbranching main axons shown in Figure 4 is likely also a key contributor to the inhibition of calcium signals in branching striatal axons tested in Figure 5A-C.

Although we believe that propagation is important condition for the results shown in Figure 4, we agree that branch point failure is not the only mechanism and have edited our manuscript to clarify this point. In particular, we hypothesize that as action potentials propagate throughout the axon, the effect of GABA-A receptors in shortening the amplitude of spikes becomes progressively greater with distance. The shorter amplitude spikes will likely reduce calcium influx and inhibit dopamine release. In addition, we hypothesize that in some cases these shorter spikes would eventually lead to an increase in branch point failures. Future experiments should seek to use whole-cell axonal recordings to directly measure action potential waveforms in the striatal axon following GABA-A activation. Discussion of this topic has been added to the main text (subsection “Mechanism of axonal GABA-A receptor mediated inhibition of striatal dopamine release”).